# A New Water Environmental Load and Allocation Modeling Framework at the Medium–Large Basin Scale

**Qiankun Liu [1], Jingang Jiang [2], Changwei Jing [3], Zhong Liu [4],\* and Jiaguo Qi [5],\***

[1]   Ocean College, Zhejiang University, Zhoushan 310058, China; 11534012@zju.edu.cn
[2]   Institute of Technical Biology & Agriculture Engineering, Hefei Institutes of Physical Science, Chinese Academy of Sciences, Hefei 230031, China; gangzg@ipp.ac.cn
[3]   School of Science, Hangzhou Normal University, Hangzhou 310018, China; changweij@hznu.edu.cn
[4]   Ningbo Scientific Research and Design Institute of Environmental Protection, Ningbo 315000, China
[5]   Center for Global Change and Earth Observation, Michigan State University, 1405 South Harrison Road, East Lansing, MI 48824, USA
\*   Correspondence: lqk19890627@163.com (Z.L.); qi@msu.edu (J.Q.);
    Tel.: +86-139-578-72922 (Z.L.); +86-136-9327-8045 (J.Q.)

**Abstract:** Waste load allocation (WLA), as a well-known total pollutant control strategy, is designed to distribute pollution responsibilities among polluters to alleviate environmental problems, but the current policy is unfair and limited to single scale or single pollution types. In this paper, a new, alternative, multi-scale, and multi-pollution WLA modeling framework was developed, with a goal of producing optimal and fair allocation quotas at multiple scales. The new WLA modeling framework integrates multi-constrained environmental Gini coefficients (EGCs) and Delphi-analytic hierarchy process (Delphi-AHP) optimization models to achieve the stated goal. The new WLA modeling framework was applied in a case study in the Xian-jiang watershed in Zhejiang Province, China, in order to test its validity and usefulness. The results, in comparison with existing practices by the local governments, suggest that the simulated pollutant load quota at the watershed scale is much fairer than the existing policies and even has some environmental economic benefits at the pollutant source scale. As the new WLA is a process-based modeling framework, it should be possible to adopt this approach in other similar geographic areas.

**Keywords:** total water pollutant control; pollutant load allocation; equity and efficiency; regional and site-specific scale; environmental Gini coefficient models; Delphi-analytic hierarchy process models

## 1. Introduction

Water pollution may cause serious environmental concerns, such as oxygen deficiency and toxic algal blooms, that are unsafe and unsuitable for daily life, industry, and even agriculture [1]. Total water pollutant control is an important policy instrument to constrain pollutant discharges in order to ensure water quality [2–4]. Proper allocation of discharge quotas or waste load allocation (WLA) based on a rational and fair basis is critical for water pollution control [5–7]. However, challenges exist in practice as WLA is directly related to the economic benefit allotment and coordination among different pollution contributors. Economic development, which is viewed as a major cause of water pollution, and environmental protection, as well as social benefits, must be balanced in principal when allocating pollution quotas [8] through a fair and rational WLA system.

The United States of America was one of the first countries to adopt a total water pollutant control policy and developed the technical guidelines for WLA in 1972 [9]. These guidelines were based on

various comparative analyses to determine optimal practical procedures that specifically consider cost, feasibility and effectiveness. Ronald et al. [10], for example, compared and evaluated eight different optimization formulations from 25 WLA methods and then eliminated one of them due to its excessive costs and overly conservative load estimates. The WLA concept was introduced to China in the 1980s and subsequently attracted considerable interest from many scholars and decision makers to assess its validity and practicality, and curtail China's pressing water pollution issues [11,12]. At the moment, WLA guidelines are applied at either region or site-specific scales [13–16], with a primary focus on single pollutant industrial point source (PS) pollution [17,18].

A notable drawback of existing WLA, when applied at town or village scales in China, is its inability to practically identify individual polluters, the terminal implementers of WLA (Figure 1). This is because the current WLA is pollutant-specific, without specific targeted sources of pollutions that can, in fact, come from many sources such as domestic sewage, agricultural non-point sources (NPS), and large-scale livestock breeding farms [19]. This ambiguity creates unfair and unpractical WLA policies, leading to ineffective practices. Therefore, there is a clear need to re-think the existing WLA and devise alternative fair allocation guidelines that can integrate multiple sources (either NPS or PS) at multiple scales, in order to effectively reduce water pollutions.

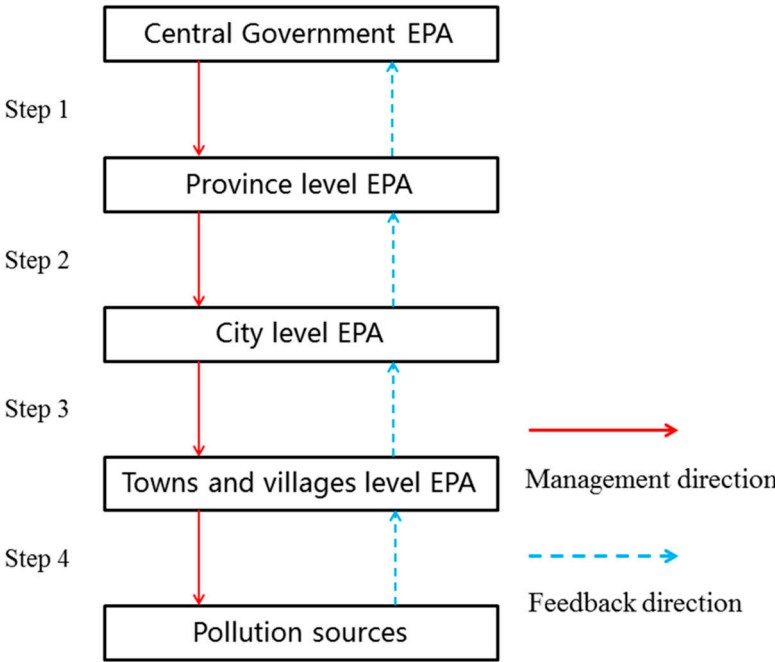

**Figure 1.** The cascade pollution discharge management structure in China: The solid red lines are the management decision making and the blue dashed lines are the cumulative feedback.

Another issue, when applied at regional scale, is related to the diverse nature of pollution contributors. Different administrative units and polluters design their own implementation plans based on their economic situation, natural resource availability, and administrative structures [6]. These units are likely to have different levels of pollution contributions [20], but the existing regional WLA policy treats them with equal pollution responsibility.

Similarly, unfairness exists at the pollution source scale, due to the nature of the pollutions (for example, industrial factories vs. residential vs. confined feeding) and the heterogeneity in economic status, social benefits, and organizational structures. These differences and their socioeconomic characteristics should be considered in order to design an effective WLA system [21]. Few WLA frameworks, to our knowledge, take into account of these diversity and heterogeneity issues among pollution sources, leading to unfair and, thus, inefficient and ineffective pollution reduction policies.

In this study, a new multi-scale and multi-sector optimal WLA framework for water pollution control was developed by integrating multi-constraint environmental Gini coefficients (EGCs) [22] and Delphi-analytic hierarchy process (Delphi-AHP) [23] models to account for the above unfairness issues. The new cascade WLA approach simultaneously allocates waste load reduction quotas at both the regional scale and site-specific scales. It addresses fairness issues and considers multiple pollution sources to account for socioeconomic benefits. The new WLA method was tested and validated in a case study in the Xian-jiang watershed, one of the most seriously water-polluted watersheds within the Yangtze River Delta, China. The results suggested that the new WLA approach provided a fair and optimal pollution discharge strategy for each pollution source in each district, implying that water quality targets can be practically achieved through policy interventions.

## 2. Multi-Scale WLA Optimization Modeling

The technical approach to integrate optimization procedures and multi-constraint criteria with socioeconomic variables is illustrated in Figure 2. Independent from the geographical characteristics (e.g., local climate, topography, and soil properties), the new WLA modeling framework integrates all pollution sources and the socioeconomic status in a watershed. Therefore, it has a broad applicability in other watersheds. The regional scale WLA (upper box in Figure 2) first uses population, GDP (gross domestic product), and total land use area data as inputs to the EGC model for allocating pollution load quotas at the administrative level (county or district level, for example). Then, the administrative-level WLA allocations are cascaded to a pollution source or a site-specific level (the lower box in Figure 2), which considers in situ discharges, population, costs associated with pollution reduction, the level of technical challenges, and pollutant discharge per unit GDP or income in the Delphi-AHP model. This model finally allocates quotas to different sectors (industrial plants, agriculture, livestock farms, and residential sewage). This integrated modeling framework was coded and implemented in the MATLAB (R2012b, ver. 8.0) environment.

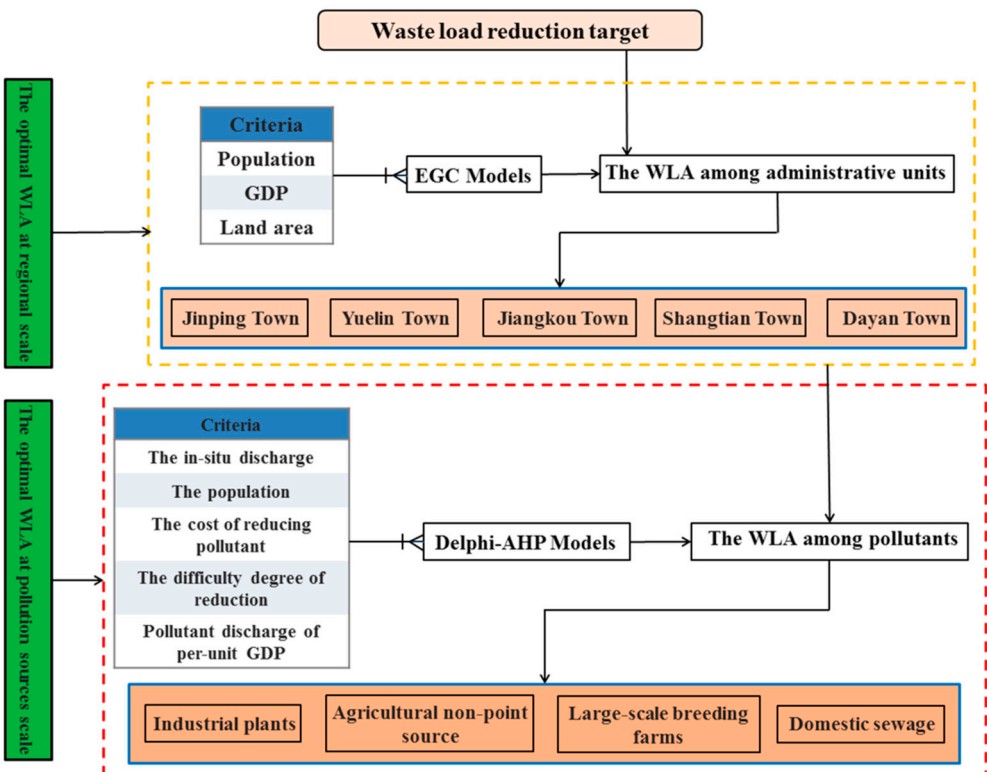

**Figure 2.** An integrated, multi-scale, and multi-sector optimal waste load allocation (WLA) framework in the Xian-jiang watershed, China.

*2.1. Regional-Scale EGC Modeling*

2.1.1. Gini Coefficient

The Gini coefficient, developed by the Italian economist Gini in the early 20th century, is an indicator to measure the extent of equality in wealth distribution and is calculated based on the Lorenz curve, using the trapezoidal area method [24,25]. The lower the Gini coefficient, the more allocation equality the society has; conversely, lower equality is indicated by a higher Gini coefficient.

2.1.2. Gini Coefficient in Environment

In recent years, the Gini coefficient has been adopted in the field of environmental inequality. From the environmental viewpoint, pollutant discharge quotas as a public issue can take advantage of the spirit of the Gini coefficient to express the fairness of various regions' emission intensities, which emphasizes the equal right in sharing the pollution discharge quota among implementers. Therefore, the EGC model was introduced to measure the inequality in the allocation of this 'resource' by comparing neighborhoods on a regional basis [26,27]. Furthermore, the EGC expands the original Gini coefficient from a single-criterion basis to a multi-criteria system. A lower EGC value suggests a more equal allocation of water pollution loads, or means that the distribution is appropriate for the region's actual social and economic development [28].

The selection of appropriate evaluation indices is crucial in the regional allocation of waste loads using EGC models and regional perspectives, such as environmental resources, local economy, and social conditions, are important in WLA quota allocation. In this study, the region's total population, GDP and land area were selected to be typical control targets of EGCs based on the following considerations [13]: First, population is a social indicator. As a public resource, each person has an equal right to enjoy the water environment capacity (WEC). This means the waste load quota should be allocated proportionally to the district populations. Second, GDP, an important economic indicator, represents incomes and, thus, the financial capacity of a local district. Third, the total land area as an indicator of natural resources is both a contributor to pollutions through, for example, NPS, but also functions as a purification medium through, for example, wetlands. Further, more land area often means that the district has more potential for industrial expansion, population growth, and economic development.

2.1.3. EGC Optimization for WLA

The optimization of WLA using EGCs at the regional scale can be realized by the following steps:

1. Determine the total amount of pollution load that has exceeded the WEC and that needs to be allocated within the watershed to meet the specified water quality goals and ensure sound water functions.
2. Compute the EGCs of the three indicators (population, GDP, and land area) under the current waste load discharge in each district as the initial values of the optimization (Equation (1)).

$$G_{0(j)} = 1 - \sum_{i=1}^{n} \left( X_{j(i)} - X_{j(i-1)} \right) \left( Y_{0(i)} + Y_{0(i-1)} \right) \tag{1}$$

where $G_{0(j)}$ is the initial EGC of criterion j; $X_{j(i)}$ is the cumulative percentage of criterion j of the ith district; $Y_{0(i)}$ is the cumulative percentage of current pollution discharge of the ith district; and n is the number of administrative units within the watershed.

3. Compute the EGCs of the three indicators (population, GDP, and land area) under the current waste load discharge in each district as the initial values of the optimization (Equation (2)).

$$
\begin{cases}
G_{(j)} = 1 - \sum_{i=1}^{n} \left(X_{j(i)} - X_{j(i-1)}\right)\left(Y_{(i)} + Y_{(i-1)}\right) \\
Y_{(i)} = Y_{0(i)} - w_i
\end{cases}
\tag{2}
$$

where $G_{(j)}$ represents the EGC of criterion j after reduction; $Y_{(i)}$ represents the cumulative percentage of waste load discharge after reduction in the ith district; and $w_i$ represents pollution load removals of the ith district.

The multi-constrained minimum EGC models can be implemented to obtain the optimal WLA scheme, i.e., to find a set of $w_i$ optimal solutions based on fairness. The optimization process can be described by the optimization objective function, as follows:

$$
\text{Min } F = \sum_{j=1}^{3} G_j
\tag{3}
$$

which represents the minimum summation of EGC$_S$ for each criterion j, and by the constraint function of the total amount of removals, which is expressed as follows:

$$
W = \sum_{i=1}^{n} w_i
\tag{4}
$$

The sum of the reduced pollution loads in each district should be the total removals (W) in the whole basin.

The constraint function for each EGC is as follows:

$$
G_{(j)} \leq G_{0(j)}
\tag{5}
$$

The EGC for every criterion j should get smaller or be at least equal to the initial EGC so that more equity can be achieved after optimization.

The constraint of the removal rate in each district is stated as follows:

$$
\begin{cases}
P_i = \dfrac{w_i}{w_{0(i)}} \\
P_{i0} \leq P_i \leq P_{i1}
\end{cases}
\tag{6}
$$

where $P_i$ is the pollution load removal rate for the ith district; $P_{i0}$ and $P_{i1}$ are the lower and upper limits of the removal rate for the ith district, respectively; and $w_{0(i)}$ is the current waste load discharge of the ith district. An appropriate upper limitation of 20% and lower limitation of 1% were set in this study to make sure each district had a waste load removal, but not one that was beyond its capability.

The multi-constrained optimization model equations can be solved using the Monte Carlo simulation method [29] to find the optimal solution vector, $w_i$, for WLA at the regional scale and to avoid causing conflicts of interest among districts.

### 2.2. Multi-Sector Scale: The Delphi-AHP Models

The AHP [23] model, an important approach to multi-criteria decision making (MCDM), can help decision-makers address a complex problem, while permitting the prioritization of alternatives. It involves four basic steps, as follows: (a) modelling a complex problem as a hierarchy, (b) the valuation of weights for each criterion, (c) the aggregation of weights into overall priorities for the alternatives, and (d) consistency test [23]. Since its introduction, the AHP has been widely used in diverse fields, such as manufacturing systems, supplier selection, strategy selection, and many others [30–33].

The hierarchical modelling of the problem and the ability to simultaneously adopt qualitative and quantitative judgements are its major strengths [23], making it appropriate for our study.

### 2.2.1. Hierarchical Structure Construction

Considering the overall benefits of environmental, economic, and social factors, as well as the technological level in the districts, a hierarchical structure of the problem was designed based on the evaluation criteria system, combining qualitative and quantitative analyses (Figure 3). The need to allocate the total pollutant discharge was selected as the overall goal (Level 1). The indicators selected at level 2 include the in situ discharge ($b_1$), the population size ($b_2$), the pollutant reduction cost ($b_3$), the technical difficulties of pollutant reduction ($b_4$), and the discharge per unit of GDP ($b_5$) for each pollution source.

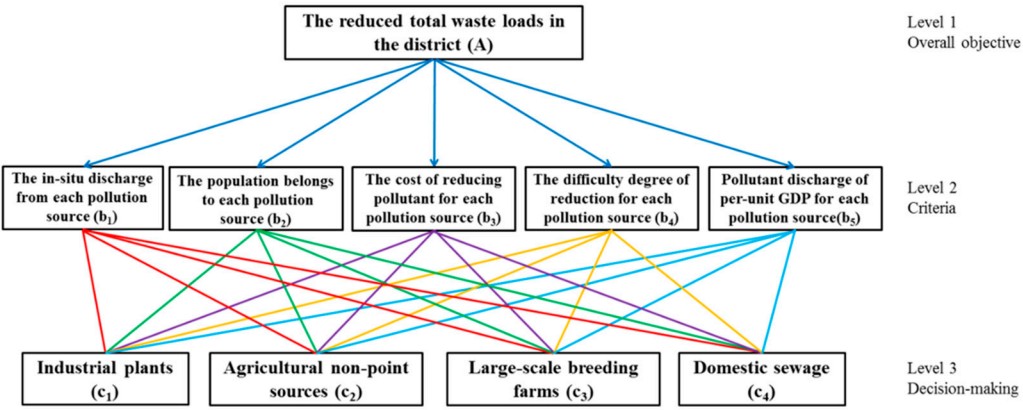

**Figure 3.** The hierarchy analysis framework of optimization criteria.

The main considerations at Level 2 are as follows: (1) The value $b_1$ not only reflects the current status of pollution source discharge, but also indicates the shares from various sectors of total pollution in a district; (2) $b_2$, as a typical social criterion, was chosen to represent the demographic distribution among different pollution sources; (3) $b_3$ and $b_4$ are important criteria of economic efficiency in the WLA between various sectors (here, we argue that WLA should be inclined to sectors with the features of easy operation and low cost in waste load reduction); and (4) $b_5$ reflects the differences in technical management levels, which can effectively promote technological innovation in pollution reduction, production efficiency, and low emissions. The main pollutant sources, as WLA receptors in the district, constitute level 3 as the final decision-makers. This optimization is to ensure that the removal quotas were allocated in an equitable and highly economically efficient way, which is very important in water resource management in China.

### 2.2.2. Pairwise Comparison Matrix Construction

The construction of the pairwise comparison, $A = \left(a_{ij}\right)_{n \times n}$, as a positive reciprocal matrix is as follows:

$$A = \begin{bmatrix} 1 & a_{12} & \cdots & a_{1n} \\ a_{21} & \cdots & a_{ij} & \cdots \\ \cdots & a_{ji} & \cdots & \cdots \\ a_{n1} & \cdots & \cdots & 1 \end{bmatrix}_{n \times n} \tag{7}$$

where $a_{ij}$ is the comparison value between object i and j; $a_{ij} = 1/a_{ji}$, $a_{ii} = 1$.

The value $a_{ij}$ estimates the relative importance among pairs of i and j at a given level using a linear scale of the integers 1–9 with their reciprocals, as advanced by Saaty [34] (Table 1), which was regarded as the best scale to represent weight ratios. To minimize the impact of individual subjectivity, the Delphi method, which has been widely used, especially for natural science and technology fields [35–37], was

used in building the above matrix. According to the expert selection principle, we adopted three steps in this study, as follows: First, we selected an evaluation team of 15 local experts in the field of water environment, who are knowledgeable about the studied watershed. The team was composed of five experts with PhD degrees, and ten engineers from environmental protection authorities in Ningbo city. Second, we asked the experts fill out a table that was designed based on the grading standard of Table 1 to evaluate the importance of mutual indicators. Finally, the survey results of all the experts are integrated in the pairwise comparisons using the geometric mean method. An analysis of the expert's opinions allowed us to minimize the individual subjectivity effects in the process of qualitative evaluation [38].

**Table 1.** The 1 to 9 fundamental scale.

| Intensity of Importance | Definition |
|:---:|:---:|
| 1 | Equal importance |
| 2 | Weak |
| 3 | Moderate importance |
| 4 | Moderate plus |
| 5 | Strong importance |
| 6 | Strong plus |
| 7 | Very strong or demonstrated importance |
| 8 | Very, very strong |
| 9 | Extreme importance |

Suppose the pairwise comparison contains n elements and p experts, then $a_{ij}$ can be calculated using the following:

$$a_{ij} = \left( \prod_{r=1}^{p} a_{ij}^{r} \right)^{\frac{1}{p}},$$

(8)

where $a_{ij}^{r}$ represents the importance of element i relative to j, determined by the rth expert.

### 2.2.3. Weight Ratios Calculation

To construct the comparison matrix $B = \left( b_{ij} \right)_{5 \times 5}$, the priorities $\vec{P} = \begin{bmatrix} p_1 & p_2 & \cdots & p_5 \end{bmatrix}^T$, which reflect the mutual importance of all indicators at the criteria level, can be derived from the eigenvalue method described in Equation (9), as follows:

$$\begin{cases} \overline{M_i} = \sqrt[5]{\prod_{j=1}^{5} b_{ij}} \\ P_i = \dfrac{\overline{M_i}}{\sum_{i=1}^{5} \overline{M_i}} \end{cases} \quad (i = 1, 2, \cdots, 5; \ j = 1, 2, \cdots 5)$$

(9)

where $\overline{M_i}$ is the geometric mean of the ith line in the pairwise matrix B and $P_i$ is the weight of the ith criterion in level 2.

The local comparison matrices $\left( C_n = \left[ \left( c_{ij} \right)_{4 \times 4} \right]_n, \ n = 1, 2, \cdots, 5 \right)$ at the decision-making level were constructed with respect to each element in the preceding level. The local priorities of $L_i = \begin{bmatrix} \vec{L_1} & \vec{L_2} & \cdots & \vec{L_5} \end{bmatrix}_{4 \times 5}$ can be calculated according to the method proposed above. Finally, with an additive aggregation, with normalization, the sum of the local priorities to unity was adopted to determine the global priority, $\vec{W} = \begin{bmatrix} w_1 & w_2 & w_3 & w_4 \end{bmatrix}^T$, at the decision-making level for the WLA among pollution sources, expressed by Equation (10), as follows:

$$\vec{W} = \begin{bmatrix} \vec{L_1} & \vec{L_2} & \cdots & \vec{L_5} \end{bmatrix} \times \vec{P}$$

(10)

where $\vec{W}$ represents the global priorities of the pollution sources; $L_i$ is the local priorities of the pollution sources with respect to the ith criteria; and $\vec{P}$ represents the weights of the criteria in level 2.

### 2.2.4. Consistency Test

As priorities make sense only if derived from consistent or near consistent matrices, the evaluation of each matrix must go through consistency verification to ensure the preferable credibility of the results. Saaty [39] proposed a consistency index (CI) (Equation (11)), which is related to the eigenvalue method, as follows:

$$CI = \frac{\lambda_{max} - n}{n - 1} \tag{11}$$

where n = the dimension of a comparison matrix; and $\lambda_{max}$ = the maximal eigenvalue of the matrix.

Having obtained the CI, it was then substituted into Equation (12) to calculate the consistency ratio (CR), as follows:

$$CR = \frac{CI}{RI} \tag{12}$$

where RI is the random index (the average CI of 500 randomly filled matrices), which is depicted in Table 2 [40].

**Table 2.** Random indices form.

| n | 1 | 2 | 3 | 4 | 5 | 6 | 7 | 8 | 9 |
|---|---|---|---|---|---|---|---|---|---|
| RI | 0 | 0 | 0.58 | 0.89 | 1.12 | 1.26 | 1.36 | 1.41 | 1.46 |

A perfect consistency should not be expected when working with the AHP [18]. When RI < 0.1, the consistency of the comparison matrix is satisfied or the priorities should be slightly revised.

## 3. Study Area

The Xian-jiang watershed lies in Ningbo city in the Zhejiang province, China, and is a typical coastal watershed within the Yangtze River Delta (Figure 4). It has a drainage area of 306.70 km$^2$ with main rivers and 5 town-level administrative divisions (Figure 4). The watershed has experienced serious water pollution problems [41,42], resulting from industrial plants, domestic sewage, agricultural run-off, and large-scale livestock breeding. The primary pollutants are chemical oxygen demand (COD), ammonia nitrogen (NH$_3$-N), and total phosphorus (TP), providing a good case study for multi-scale and multi-sector WLA analysis.

This watershed was set to reach a water quality of grade II and III for the upper and lower reaches, respectively, by 2020, according to the standards set by Chinese EPA [43] as part of the 13th National Five-Year Plan of China. In this study, a steady-state 1-D water quality response model coupled with the matrix algorithm and the section control method was integrated within the WLA framework to calculate the pollutants that need to be reduced in order to meet the stated targets.

The watershed was divided into 1350 uniform computational elements for model calculation. The flow velocity in each element was obtained using the Manning hydrodynamic equation [44], and then the concentration of a specific pollutant at *x* distance was calculated based on the water balance equation [45], expressed as follows in Equation (13):

$$\frac{\partial c}{\partial t} + u\frac{\partial c}{\partial x} = E\frac{\partial^2 c}{\partial x^2} - k_1 c \tag{13}$$

where c is the pollutant concentration (mg/L); u is the flow velocity (m/s); t is the time of river water flowing through from the headwater to somewhere (s); x is the distance that river water flows through in time t (m), x(t) = ut and $k_1$ is the pollutant degradation coefficient (d$^{-1}$); and E is the longitudinal dispersion coefficient.

The computed pollutant concentrations of each element were applied to the section control method (Equations (14)) to assess the water environmental capacity of each element.

$$W_i = Q_i(C_{Si} - C_{0i}) + Q_i C_{Si}(e^{k_i t_i} - 1) + \sum_{l=1}^{n} q_l C_{si} + C_{si} \sum_{l=1}^{n} q_l(e^{k_i t_i} - 1) \tag{14}$$

where $W_i$, $C_{si}$, and $C_{0i}$ represent the water environmental capacity, the target pollutant concentration, and the actual concentration of the pollutant of element i, respectively; $k_i$ is the degradation coefficient of the pollutant in element i; and $t_i$ is the time period used to flow through element i.

Eventually, the pollution loads that need to be reduced are 340.16, 25.11, and 11.41 tons for COD, $NH_3$–N, and TP, respectively, in order to meet the stated water quality targets by year 2020. Detailed descriptions of the simulation process and model validation are available in Liu et al. [19], and are not presented here for the sake of brevity.

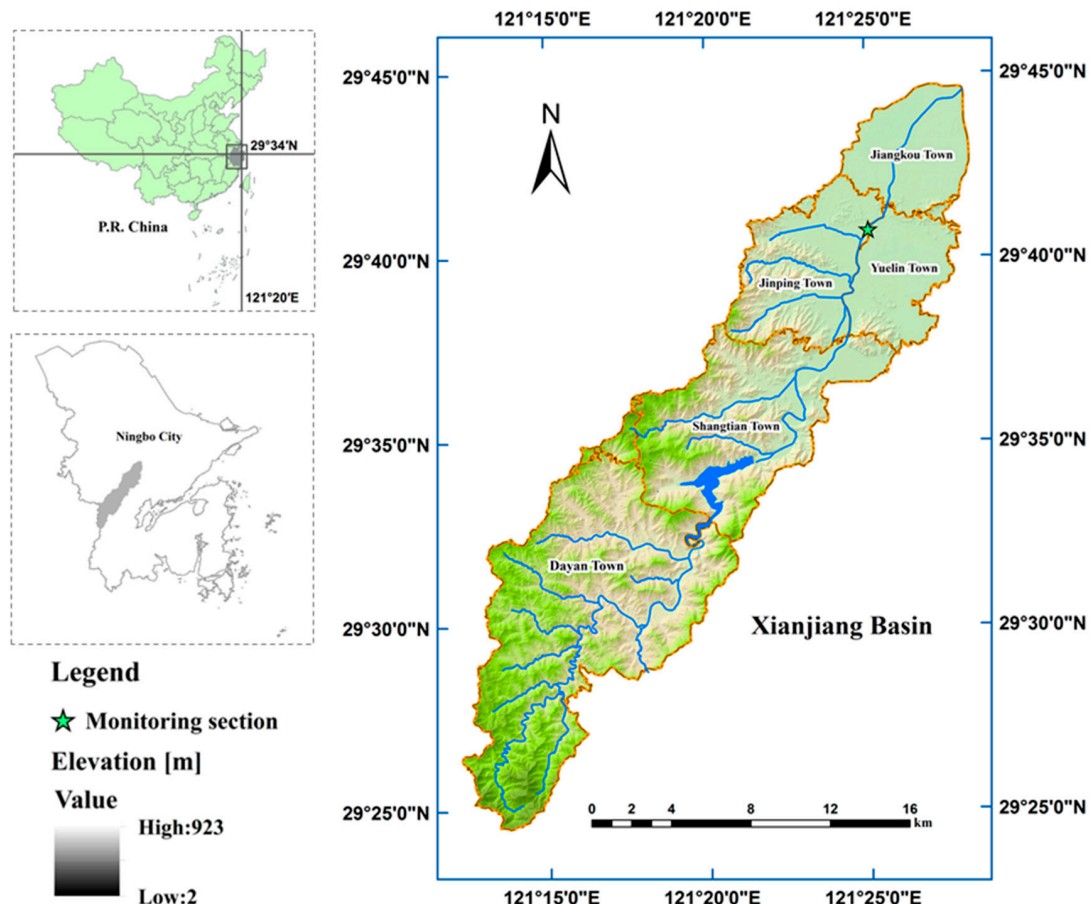

**Figure 4.** The geographic characteristics of Xian-jiang watershed within Ningbo City, China.

## 4. Results and Discussion

### 4.1. Regional-Scale Allocation Results

#### 4.1.1. Optimal WLA Results in Districts

The population, GDP, and land area data of each town in the Xian-jiang watershed were obtained from the Towns Agency of Statistics of Ningbo City (2015) [46], while the in situ discharges of pollutants (2015) for each town in the basin were calculated (Table 3). The EGC for each criterion under the

constraints was optimized, and the sum of all criteria's EGCs was minimized to obtain an optimal solution vector; namely, the reduced waste loads of the five towns considered.

**Table 3.** The criteria statistics and in situ discharges of pollutants of districts in the Xian-jiang watershed (2015). COD, chemical oxygen demand; NH$_3$-N, ammonia nitrogen; TP, total phosphorus; GDP, gross domestic product.

| Districts | Criteria | | | Pollutants In Situ Discharge (t) | | |
| --- | --- | --- | --- | --- | --- | --- |
| | Population | GDP (Ten Million Chinese Yuan) | Land Area (km$^2$) | COD | NH$_3$-N | TP |
| Jinping Town | 112,209 | 891.94 | 41.82 | 2664.71 | 201.41 | 50.68 |
| Yuelin Town | 49,262 | 1131.02 | 27.09 | 1696.88 | 97.93 | 29.66 |
| Dayan Town | 13,591 | 133.28 | 127.53 | 393.79 | 28.83 | 11.76 |
| Jiangkou Town | 29,885 | 367.74 | 31.10 | 1599.39 | 128.92 | 25.94 |
| Shangtian Town | 19,071 | 234.46 | 73.53 | 411.91 | 38.07 | 12.73 |

As shown in Figure 5, the targeted pollutant removals and proportion of COD discharge in Jiangkou, Jinping, Yuelin, Dayan, and Shangtian were 198.09 t (12.39%), >72.91 t (2.74%), >43.25 t (2.55%), >15.91 t (4.04%), and >10.00 t (2.43%), respectively. Jiangkou (14.96 t) and Jinping (5.49 t) were the two districts with the largest NH$_3$-N reduction loads, which accounted for 11.61% and 2.73% of the total removal rates, respectively. A total of 11.41 t of TP pollutants needed to be reduced, of which the largest removal (proportion) was in Jiangkou, with 5.24 t and 20.22%, followed by Jinping and Dayan with 3.97 t (7.84%) and 0.63 t (5.36%), respectively. The removal rate of the remaining districts (Shangtian and Yuelin) were relatively low, and were both less than 4.00%, with removals of 0.50 t and 1.06 t, respectively.

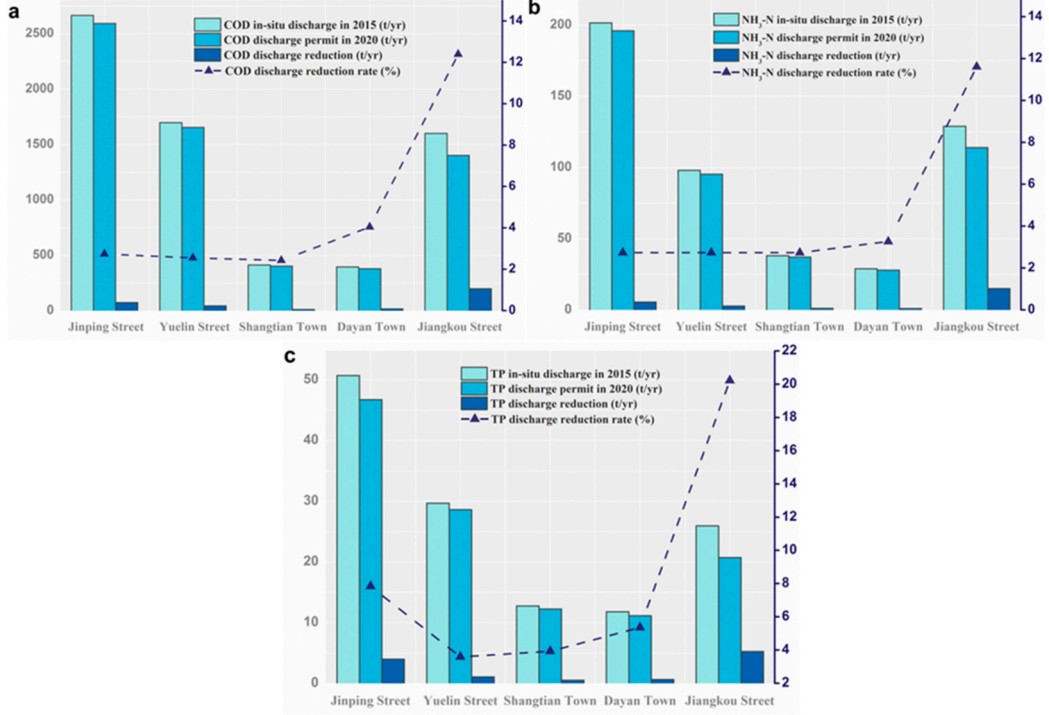

**Figure 5.** The in situ discharge (2015) and allocation of pollutant discharge quotas after optimization in the Xian-jiang watershed: (**a**) chemical oxygen demand (COD); (**b**) ammonia nitrogen (NH$_3$-N); and (**c**) total phosphorus (TP).

The regional allocation results revealed that Jiangkou was significantly higher than other districts in terms of both load removals and proportion of the three pollutants. Part of the reason was the high

in situ pollutant discharges in the district and part was the modes of social economic developments. Conversely, as the town with the lowest pollutant discharge per unit of GDP and population, Shangtian was granted a lesser quota both in removal and rate, as predicted by the model.

Interestingly, the towns with the largest pollutant discharges were not necessarily those with the highest proportions of removal quota because the allocation of a pollutant discharge quota at the regional scale considered all factors, including the region's economic efficiency and social equality, not just the magnitude of the in situ discharge of pollutants. For example, although Jinping and Yuelin exceeded the in situ COD discharges of Jiangkou, it was assigned a lesser pollutant removal and proportion. This is probably due to the backward modes of social economic development in Jiangkou, such as less-developed domestic sewage networks and large areas of extensive agriculture, which resulted in higher discharges of pollutants per unit of GDP and population, compared with the other two towns.

### 4.1.2. EGCs Before and After Optimization

The EGC optimization models were used to calculate the EGC of each criterion among the districts (Table 4) and draw the Lorenz curves before and after the optimization for the three pollutants.

**Table 4.** $EGC_S$ of multiple criteria in the Xian-jiang watershed.

| Pollutants | Criteria | EGC (P [1]) | EGC (G [1]) | EGC (L [1]) | Total |
|---|---|---|---|---|---|
| COD | Before optimization | 0.162 | 0.215 | 0.583 | 0.960 |
| | After optimization | 0.152 | 0.201 | 0.576 | 0.929 |
| | Decrease | 0.010 | 0.014 | 0.007 | 0.031 |
| $NH_3$-N | Before optimization | 0.146 | 0.271 | 0.569 | 0.986 |
| | After optimization | 0.129 | 0.258 | 0.569 | 0.956 |
| | Decrease | 0.017 | 0.013 | 0.000 | 0.030 |
| TP | Before optimization | 0.141 | 0.217 | 0.521 | 0.879 |
| | After optimization | 0.126 | 0.197 | 0.519 | 0.842 |
| | Decrease | 0.015 | 0.020 | 0.002 | 0.037 |

[1] P, G, and L indicate the abbreviations for population, GDP, and land area, respectively. EGC, environmental Gini coefficient.

Take the COD as an example, the EGC of population vs. COD in situ discharge is the smallest, at 0.162, and is within the reasonable range in WLA equity at the regional scale, indicating that the current distribution of COD discharge among districts is balanced according to population. The GDP-based EGC for COD in situ discharge among districts is 0.215, which keeps the COD discharge to the districts relatively reasonable in relation to local economy, but still has the potential for optimization. In particular, the greatest distribution inequity of COD discharge at regional scale occurs in the land area, with an EGC of up to 0.583, exceeding the warning sign of equity (0.400) [6], suggesting that the current COD load discharge in the five districts does not match well with the land area indicator.

Interestingly, considering the geographical pattern of pollution sources and function regionalization in the watershed using ArcGIS (ver. 10.2) (Figure 6), it can be further inferred that the high EGCs corresponding to land area are mainly imputed to Dayan town. This accounts for nearly half of the total land area (41.95%), but accommodates only little pollution discharge from pollution sources, making up only 5.82% (COD), 5.82% ($NH_3$-N), and 8.99% (TP) of the total pollutant discharges in the entire watershed. The situations of the other pollutants are similar to that of COD, having the same distribution pattern of EGCs for the equivalent criteria (Table 4).

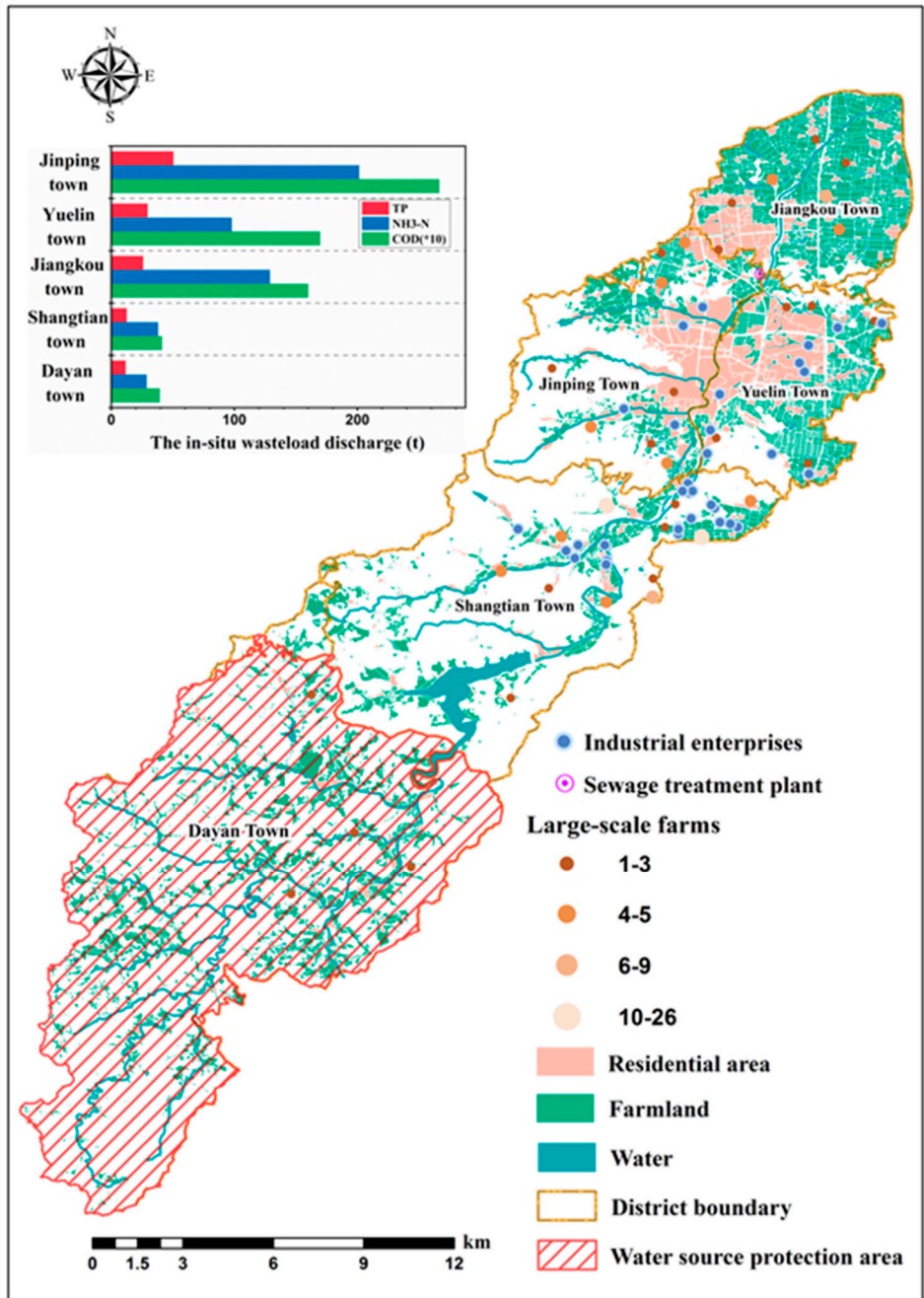

**Figure 6.** The administrative divisions, pollution source distribution, and geographical patterns of the Xian-jiang watershed.

Table 4 and Figure 7 highlight the decrement and amplitude of EGCs corresponding to the three criteria after optimization. Note that the EGCs of COD, NH$_3$-N, and TP after optimal WLA at the regional scale were all less than those of initial pollutant discharge, and the Lorenz curves after optimal allocation were closer to approaching the line of absolute equality. The results revealed that the optimal allocation of the removals at the regional scale brought more accordant and equitable responsibilities in relation to the districts' respective shares of socioeconomic development, and the optimal allocation coordinated the pollutant discharge quota with the natural environment. In other words, a more equitable pollutant discharge quota at the regional scale was achieved after WLA optimization using the EGC models.

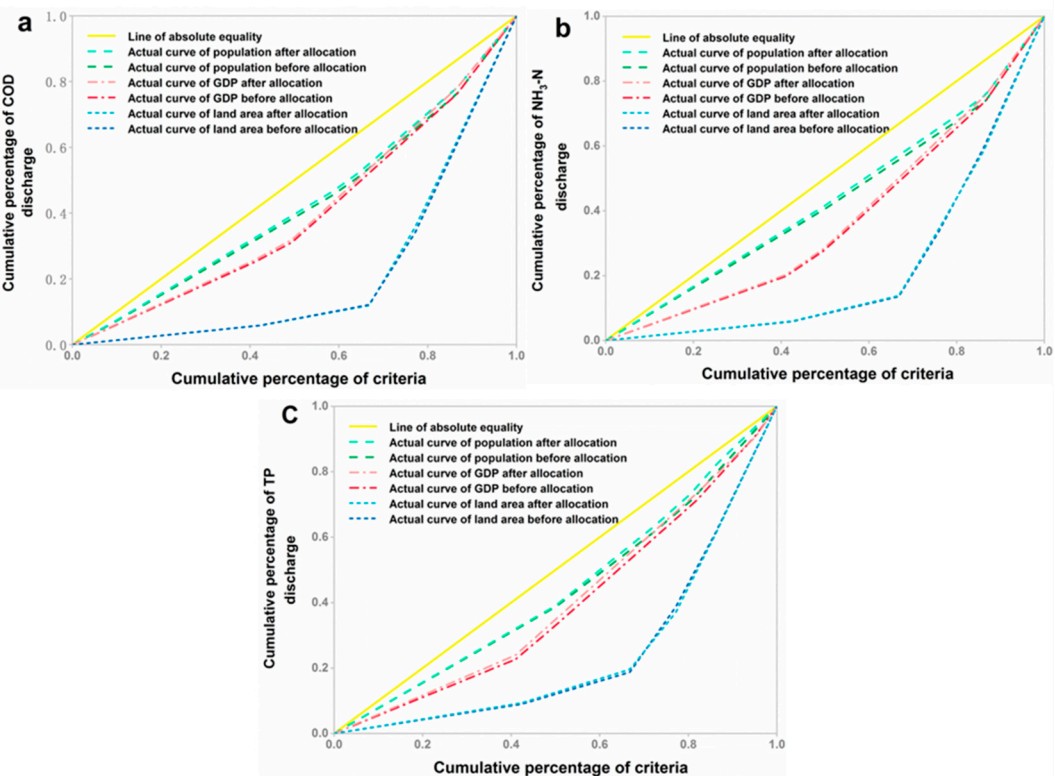

**Figure 7.** Lorenz curves of pollutant discharge based on the three criteria before and after allocation: (**a**) COD; (**b**) NH$_3$-N; and (**c**) TP.

In view of the overall trends, as depicted in Figure 7, it is noteworthy that the EGCs and the Lorenz curves do not change very much after the optimization. Two reasons can be identified as the cause, as follows: (1) The pollution discharge had been distributed to each district relatively equally according to the criteria, i.e., population and GDP, before optimization; and (2) there are upper and lower limits on the allocation of pollution removal to each district in order to be compatible with the local socioeconomic conditions. Considering the constraints in the EGC optimization models, the shape of the Lorentz curve can only be adjusted gradually to avoid an out-of-range affordability of removals for local managers, due to the tenacious struggle to decrease EGCs.

### 4.1.3. Factor Analysis

A contribution coefficient method was used to further determine the regional inequality factors referring to waste load in situ discharge vs. the criteria. The contribution coefficient, taking 1 as the threshold, is the contribution ratio of regional evaluation criteria to in situ pollutant discharge in a certain district, expressed as Equation (15) [47], as follows:

$$CC_j = \left( M_{ij}/M_j \right)/\left( W_i/W \right) \tag{15}$$

where $CC_j$ represents the population contribution coefficient (PCC), the green contribution coefficient (GCC), and the land area contribution coefficient (LACC) with respect to the criteria j of population, GDP, and land area, respectively; $M_{ij}$ is the magnitude of criterion j in the ith district, and $M_j$ is the sum of criteria j in the watershed; $W_i$ is the pollutant in situ discharge of the ith district; and W is the total pollutant discharges in the watershed.

Table 5 provides the CCs of the three criteria at each district in the watershed. As shown, in 2015, the PCCs of COD, NH$_3$-N, and TP pollutants for Jiangkou were lowest in the region, with 0.56, 0.51, and 0.67, respectively. This is possibly due to the relatively undeveloped economic and living

conditions, with some deficiencies in domestic sewage treatment systems. Moreover, the district with well-developed river systems and fairly fertile soil, an area in the plain river network of the lower Xian-jiang watershed, is the primary rice-growing area in the watershed (Figure 6). Paddy fields make up as high as 83.2% of the whole area, leading to serious agricultural NPS pollution. In short, all these factors produced an increasingly incisive contradiction between the local population and the water environment, making this an unfair-factor district in terms of EGCs vs. population.

**Table 5.** Population contribution coefficients (PCCs), green contribution coefficients (GCCs), and land area contribution coefficients (LACCs) of districts for the three pollutants in the Xian-jiang watershed (2015).

| Districts | PCC | | | GCC | | | LACC | | |
|---|---|---|---|---|---|---|---|---|---|
| | COD | NH$_3$-N | TP | COD | NH$_3$-N | TP | COD | NH$_3$-N | TP |
| Jinping Town | 1.40 | 1.23 | 1.29 | 0.82 | 0.79 | 0.83 | 0.35 | 0.34 | 0.36 |
| Yuelin Town | 0.88 | 1.11 | 0.97 | 1.64 | 2.07 | 1.81 | 0.36 | 0.45 | 0.40 |
| Dayan Town | 1.04 | 1.04 | 0.68 | 0.83 | 0.83 | 0.54 | 7.28 | 7.28 | 4.71 |
| Jiangkou Town | 0.56 | 0.51 | 0.67 | 0.56 | 0.51 | 0.67 | 0.44 | 0.40 | 0.52 |
| Shangtian Town | 1.27 | 1.11 | 0.87 | 1.40 | 1.11 | 0.87 | 4.01 | 3.18 | 2.51 |
| Mean | 1.03 | 1.00 | 0.89 | 1.05 | 1.06 | 0.94 | 2.49 | 2.33 | 1.70 |

Yuelin town, with the highest economic level in the basin, contributed 41% to the total GDP and received the largest GCCs for COD (1.64), NH$_3$-N (2.07), and TP (1.81). This indicated that the contribution rate of GDP to the entire region was more than that of pollution discharge in this district. It further revealed the advancements in the cleanliness of production processes and sewage treatment efficiency in this district. Although Yuelin exceeded two-fold the in situ pollutant discharges of Shangtian and Dayan towns, it was assigned to a lesser pollutant removal. In addition, the GCCs of Shangtian for COD and NH$_3$-N also surpassed the threshold of the green contribution factor (1.00). Moreover, the GCCs of COD and NH$_3$-N in Jiangkou, and TP in Dayan, both belonging to the less-developed areas, were less than the green contribution standard, with low values of 0.56, 0.51, and 0.54, respectively. It could be speculated that GDP output in these regions is characterized by high pollution and low efficiency, which are the main factors leading to the unfairness of WLA at the regional scale based on the GDP index.

Dayan town, referring to the criterion of land area, presented the largest LACCs among the districts for COD (7.28), NH$_3$-N (7.28), and TP (4.71), owing to its unique geographical location. In spite of vast expanses and richness in natural resources, the district is completely subject to the water source conservation area, leading to strict restrictions on local resident size and density, industrial development, and agricultural scale for the protection of drinking water security (Figure 6). In contrast, the LACCs of the three pollutants in Jinping and Yuelin towns were all below the value of 0.50, revealing a heavy discharge of pollutants into the river per unit of land area in these districts.

Different from the PCC and GCC, the LACC is not as high as it could be when there is a low utilization of land resources. Thus, the high LACCs of Dayan town should be lowered accordingly. However, our results found that its specific geographical location (water source conversation area) makes it hard for the area to be adjusted by human intervention, such as demographic migration, industrial distribution adjustment, and land use and land cover conversion. The human-induced adjustment may result in the destruction of the original ecological environment and directly threaten the safety of drinking water, which would instead lead to an 'inequity' in WLA. Consequently, the high LACCs are acceptable in this particular water function division. At the same time, we urge the optimal allocation of removals at the regional scale with the assistance of geographic information system (GIS) technology.

Furthermore, to mitigate the polluted water more efficiently, targeting a specific area instead of a whole watershed has been recommended as a cost-effective method in many previous studies [48–51]. Hence, the spatial zonation, in which the PCC and GCC were synthetically considered, into critical

source areas (CSA) (PCC < 1, GCC < 1), improving areas (PCC < 1, GCC > 1 or GCC < 1, PCC > 1), and safety areas (PCC > 1, GCC > 1) among the five districts was performed by ArcGIS (ver. 10.2) (Figure 8).

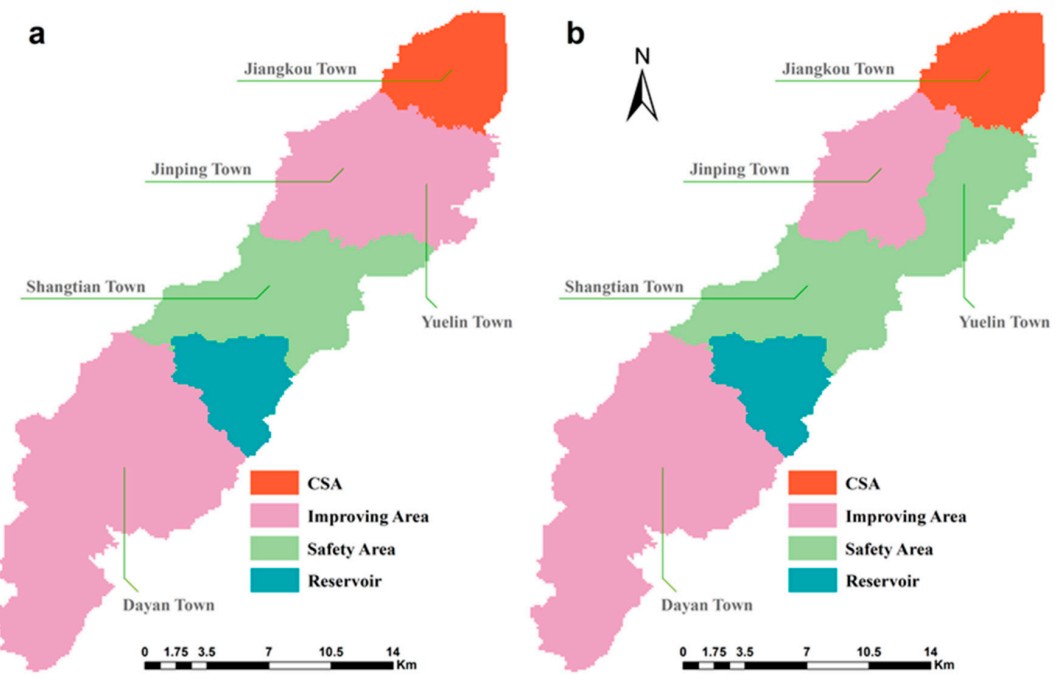

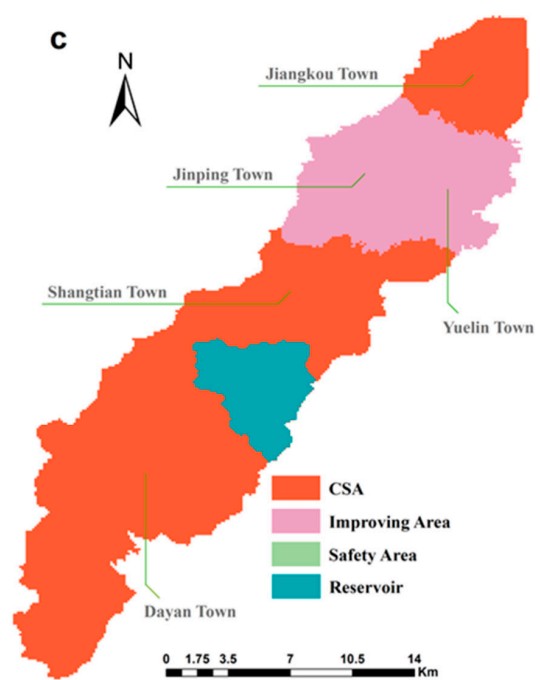

**Figure 8.** The spatial pattern of WLA unfair-factor districts: (**a**) COD; (**b**) NH$_3$-N; (**c**) TP.

As seen in Figure 8, both COD and NH$_3$-N have a similar pattern of unfair-factor districts, except Yuelin town. Among them, only Jiangkou town was identified as a CSA of unfair factors in the watershed, revealing that this area faces an acute contradiction between population, economic profit, as well as the water environment under per-unit pollutant discharges, and should be managed and

controlled preferentially for the two pollutants. Hence, the EGC optimal models handed out the most discharge removals and highest proportions of the three pollutants to this district. Meanwhile, the fairness of the optimization allocation results of using EGCs models at regional level was further verified. Conversely, Shangtian town was characterized by having both PCC and GCC values of more than 1.00, and therefore belongs to the safety area, which exhibited a high-efficiency economic output with low consumption of WEC, while there was less impact of population scale on the water environment. In other words, with limited funds and manpower, environmental protection agencies can directly skip these regions in the total pollutant control process. Water environmental management in improving areas, which either have a high level of economic development but a sharp conflict between population and environment or are under a less stressful environment from per-unit population but have low environmental and economic benefits, should improve their deficiencies while maintaining the status level of superior contribution coefficients.

In particular, for TP, a significant difference was found for the grade distribution of unfair-factor districts compared with COD and $NH_3$-N, in that all the five districts were identified as CSAs or improving areas, and CSAs accounted for a higher proportion of administrative units than with the other two pollutants, suggesting that the in situ allocation of the TP discharge quota at the regional scale should preferentially be improved.

*4.2. Results of Pollutant Source Scale Allocation*

4.2.1. Pairwise Comparisons and Synthesis of the Weights

Jinping town was identified as the inner city of the watershed with multi-sector pollution, and was selected as a typical region to perform the optimal WLA at the pollution source scale for the three pollutants using the coupled Delphi-AHP algorithm. The partial parameters of the criteria at level 2 can be achieved through a quantitative calculation using the 2015 statistical yearbook data of villages and towns in Ningbo city (Table 6) [46]. Furthermore, the above-mentioned Delphi method was adopted to evaluate the indicators with quantizing difficulties ($b_3$ and $b_4$).

**Table 6.** The partial indicators of alternatives at the decision-making level, with respect to the elements of the criteria level (2015).

| Pollution Sources | In Situ Discharge of Pollutants (t) | | | Population Scale | | | Pollutant Discharge Per Unit of GDP [1] | | |
|---|---|---|---|---|---|---|---|---|---|
| | COD | NH₃-N | TP | COD | NH₃-N | TP | COD | NH₃-N | TP |
| Industrial plants | 16.15 | 0.80 | 0.00 | 12,340 | 12,340 | 0 | 1.03 | 0.05 | 0 |
| Agricultural NPS | 158.33 | 6.61 | 5.66 | 1390 | 1390 | 1390 | 25.28 | 1.06 | 0.90 |
| Large-scale breeding farms | 82.57 | 8.00 | 6.45 | 360 | 360 | 360 | 13.17 | 1.28 | 1.03 |
| Domestic sewage | 1665.89 | 159.67 | 23.53 | 82,514 | 82,514 | 82,514 | 5.61 | 0.54 | 0.08 |

[1] Pollutant discharge per-unit of GDP: kg per 10,000 yuan. NPS, non-point sources.

The pairwise comparisons at the criteria level and the decision-making level were constructed on the basis of the above-mentioned statistical results. Subsequently, the respective local priorities of the three pollutants, namely, the importance weights of the attributes in each level, were solved using the principal eigenvector method [39], which was fully illustrated in Section 2.2.3. The maximum eigenvalues and the consistency test of comparison matrices are depicted in Table 7. The consistency test confirmed that the obtained pairwise comparisons were satisfactory in relation to consistency requirements (CR < 0.1), since they revealed a small inconsistency, inducing only a small distortion.

**Table 7.** The maximum eigenvalues and consistency parameters of the three pollutants.

| Pollutants | Pairwise Comparison Matrices | Maximum Eigenvalue ($\lambda_{max}$) | Consistency Index (CI) | Consistency Ratio (CR) |
|---|---|---|---|---|
| | B | 5.057 | 0.014 | 0.013 |
| | b1-C | 4.158 | 0.053 | 0.058 |
| | b2-C | 4.165 | 0.055 | 0.061 |
| COD | b3-C | 4.051 | 0.017 | 0.019 |
| | b4-C | 4.051 | 0.017 | 0.019 |
| | b5-C | 4.135 | 0.045 | 0.050 |
| | b1-C | 4.158 | 0.053 | 0.058 |
| | b2-C | 4.165 | 0.055 | 0.061 |
| NH$_3$-N | b3-C | 4.051 | 0.017 | 0.019 |
| | b4-C | 4.051 | 0.017 | 0.019 |
| | b5-C | 4.102 | 0.034 | 0.038 |
| | b1-C | 3.029 | 0.015 | 0.025 |
| | b2-C | 3.065 | 0.032 | 0.056 |
| TP | b3-C | 3.009 | 0.005 | 0.008 |
| | b4-C | 3.009 | 0.005 | 0.008 |
| | b5-C | 3.025 | 0.012 | 0.021 |

Our preliminary results (Figure 9) revealed that the importance sequence of the five criteria at level 2 was $b_5$ (0.41), > $b_1$ (0.25), > $b_3$ (0.14) = $b_4$ (0.14), > $b_2$ (0.06). The criterion $b_5$ (pollutant discharge per unit of GDP), as the most important factor judged by experts in the field in Section 2.2.2, reflecting the environmental economic benefits of alternative pollution sources, captures 41% of the entire sum of weights, making it the most determinant criteria. This is understandable since economic development is the paramount issue that governments care about, especially for developing countries such as China, despite this factor's negative impacts on the environment, leading to the extreme importance of the tradeoff between economic development and environmental protection. The relative preference for $b_5$ can effectively promote technical innovation and guide waste load discharge in a low-level and high-efficiency direction.

Specific to each sector, taking COD as an example, domestic sewage at the decision-making level is the primary weight in influencing and shaping WLA among pollutant sources, with respect to $b_1$ (0.65) and $b_2$ (0.58) (Figure 9). This can be explained by the fact that the magnitude of domestic pollution, with respect to both the in situ pollutant discharge and the population scale, is much higher than the other sectors in the district. Agricultural NPS ($c_2$) and large-scale livestock farming ($c_3$) were identified as determinants in $b_3$ ($c_2$ with a priority of 0.47), $b_4$ ($c_3$ with a priority of 0.47), and $b_5$ ($c_2$ with a priority of 0.58).

In particular, the industrial PS discharge has the smallest weight ratios among the alternatives, in correspondence with all the criteria except $b_2$ (large-scale livestock breeding source, 0.04), with priorities of 0.04 ($b_1$), 0.07 ($b_3$), 0.07 ($b_4$), and 0.04 ($b_5$), respectively. One possible explanation is that not only does industrial PS have the lowest waste load discharge, but more crucially, it has the highest GDP output per unit of pollutant discharge, revealing a high environmental economic benefit far ahead of any other sectors. In addition, the cost and difficulty degree of pollution removal for industrial PS, specifically considering costly wastewater treatment facilities, exorbitant operating costs, and its economic contribution to local employment and taxation, were much higher compared to the other sectors, such as agricultural and large-scale farming sources.

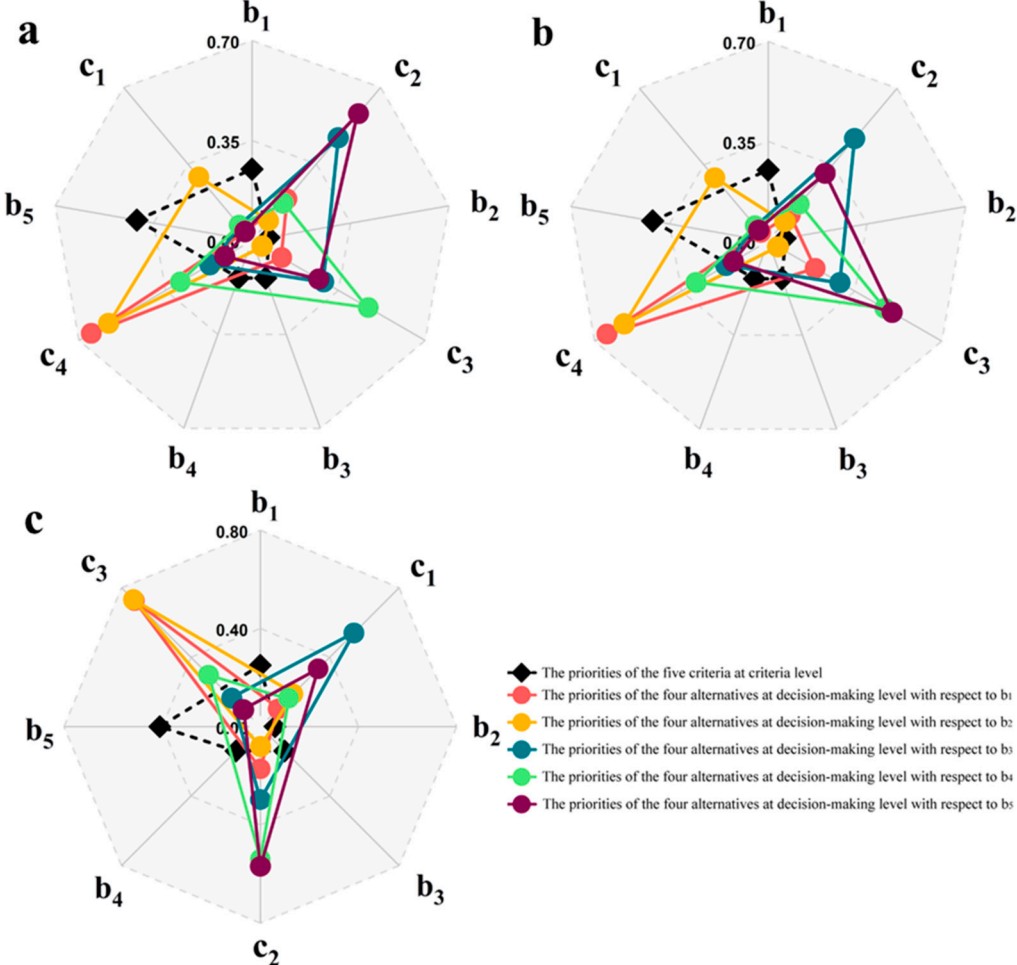

**Figure 9.** The local priorities of elements at criteria and decision-making levels: (**a**) COD; (**b**) NH$_3$-N; and (**c**) TP.

Subsequently, the results of the local priorities across all criteria were aggregated using Equation (10) to obtain the global priorities of the three pollutants, which presented an overall preference rating of WLA among each sector in the district (Table 8). The results indicated that the pollution source for COD, NH$_3$-N, and TP that possessed the highest weights was agricultural NPS (COD: 0.38) and large-scale breeding farms (NH$_3$-N, TP: 0.36 and 0.40), and the next sector was domestic sewage with priorities of 0.30 (COD), 0.32 (NH$_3$-N), and 0.33 (TP), respectively. Conversely, industrial PS was assigned with the least weight for COD (0.07) and NH$_3$-N (0.07), and agricultural NPS for TP with a priority of 0.27, referring to regional waste load removals. This implies that industrial PS was relatively unimportant in WLA at the pollutant source scale.

**Table 8.** The global priorities across the alternatives at the decision-making level, with respect to the overall objective.

| Pollutants | The Normalized Global Priorities | | | |
|---|---|---|---|---|
| | C$_1$ | C$_2$ | C$_3$ | C$_4$ |
| COD | 0.07 | 0.38 | 0.25 | 0.30 |
| NH$_3$-N | 0.07 | 0.25 | 0.36 | 0.32 |
| TP | 0.00 | 0.27 | 0.40 | 0.33 |

#### 4.2.2. The Optimal WLA among Pollution Sources

WLA was completed for the pollution sources in the district according to the global priorities at the decision-making level (Figure 10). The results revealed that agricultural NPS had the largest removal of COD, with up to 27.71 t, and the large-scale livestock breeding source was the biggest contributor of waste load removals for $NH_3$-N (1.98 t) and TP (1.59 t).

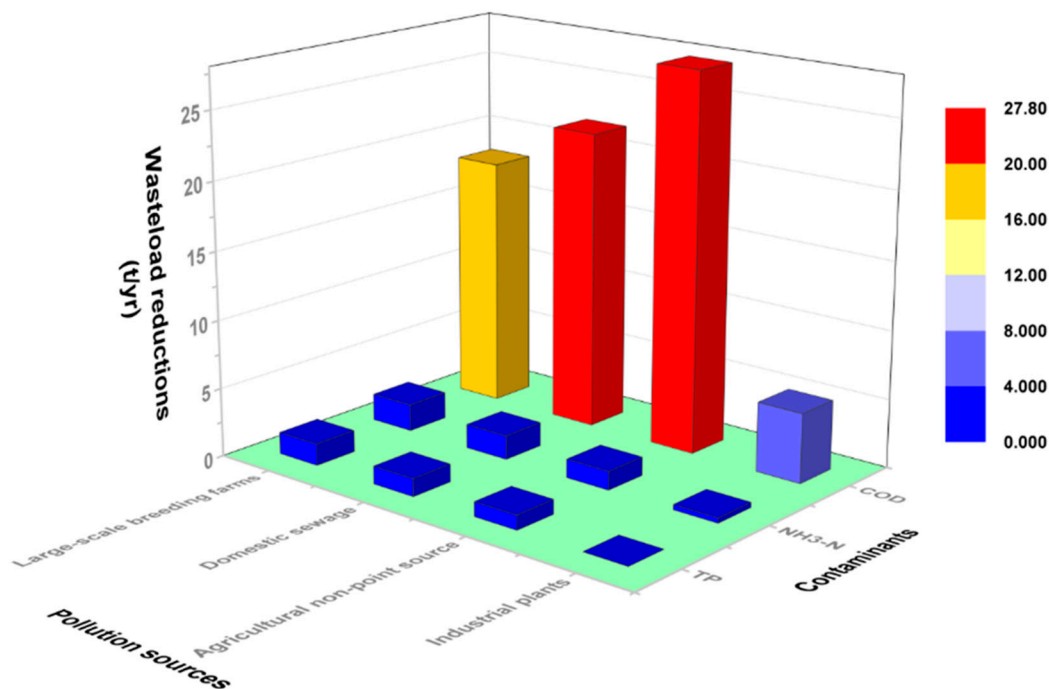

**Figure 10.** The WLA of the three pollutants among pollution sources in Jinping town.

For COD, the agricultural source had the second biggest in situ discharge, which was slightly inferior to domestic sewage, and, concurrently, had a lower cost of discharge reduction compared to the other sectors. Moreover, it was noticed that the pollutant discharge per unit of GDP (25.28 kg/ten thousand yuan) of agricultural NPS was much greater than that of other pollution sources, being 1.92, 4.51, and 24.54 times greater that of large-scale livestock breeding (13.17 kg/ten thousand yuan), domestic sewage (5.61 kg/ten thousand yuan), and industrial plants (1.03 kg/ten thousand yuan), respectively (Table 6). Such results suggest that the COD discharge from agricultural NPS not only has a significant impact on the water environment, but more seriously, it leads to economic inefficiency in water resource utilization. China has to feed 20% of the world's population with 7% of the world's cropland [52], leading to the excessive use of fertilizer and manure to increase food production. Conversely, less developed agricultural technology and extensive cultivation result in lower fertilizer application efficiency and a larger loss of nutrients [53]. Therefore, controlling COD discharge from agricultural sources has become the most critical and preferred way for environmental management sectors to achieve the goal of clean water in this district, such as through the gradual reduction of fertilizer usage, the establishment of ecological agriculture, and the control of water and soil loss.

Large-scale livestock breeding, different from COD, has the second $NH_3$-N and TP discharges in size, following domestic sewage, and has the most pollutant discharge per unit of GDP; namely, the lowest environmental and economic performance generated by per-unit pollutant discharge. On the other hand, synthetically considering its low removal difficulty and cost relative to sewage treatment facilities for industrial plants and underground sewer networks for domestic sewage, the Delphi-AHP optimal models expressed a WLA preference for the large-scale breeding source, thereby making it the key sector for removals of $NH_3$-N and TP in the district.

Industrial enterprises above the designated size were the sector with the least removal for COD and $NH_3$-N, with only 5.1 t and 0.38 t, respectively. This is mainly due to its minimal waste load discharge and, furthermore, its maximal economic performance under per-unit pollutant discharges, as compared with the other three sectors in the system. Our optimal results confirmed the viewpoint, as noted by Sun et al. [13], that the allocated sectors of equal pollution discharge with higher economic efficiency should get more shares of the waste load discharge quota.

## 5. Conclusions

Local EPAs used to allocate waste removal directly to districts or pollution sources within their jurisdictions through administrative orders, based primarily on their past experiences [54]. However, a common consequence of these WLA allocations is unfairness that leads to disputes and blaming each other for shared waste responsibilities, owing to the fuzzy allocation basis and biased subjectivity. The developed WLA framework allocates waste load removal simultaneously, at multiple scales and among different sectors, considering both the principles of equity and efficiency for the specific implementers. This is very valuable for decision-makers in providing critical information (i.e., the best compromise solutions for WLA) and practical guidance on water pollution control. The new modeling framework, based on the premise of equality, minimizes environmental costs while maximizing economic efficiency, which is extremely important for communities in developing countries.

The results revealed that the removal and proportions of pollutants are significantly associated with the region's actual socioeconomic development modes, which confirms the viewpoint that socioeconomic factors will have significant impact on water management in the future, as noted by Reynard et al. [55]. Inadequate sewage networks, the lack of wastewater treatment technologies, and intensive land cultivation resulted in high pollutant discharges per unit of GDP and population. Thus, for local authorities, these are the key targeted regions for the total waste load control. Decision-makers should encourage advanced regions in their continuous efforts to improve the economic efficiency of water environmental resources, while impelling relatively less developed areas to economically transition and gradually phase out inefficient sectors.

The distribution of unfair districts (Figure 8) provided information on how different waste load removal priorities should be considered when different districts are targeted for a specific pollutant. Meanwhile, for different pollutants, the urgency also varied throughout the administrative units across the watershed. This suggested that the current one-size-fits-all allocation strategy of waste removal adopted by environmental management sectors should be changed, and instead of preferentially reducing total pollution loads, they should focus on key pollutants (TP in our study) and regions (CSAs). This can help decision-makers save significant costs under the conditions of limited capital and energy.

The WLA results at the pollution sources scale suggest agricultural NPS as the sector with the largest removal quota of COD, as they are the large-scale breeding source for $NH_3$-N and TP. There are certain characteristics that high-reduction sectors tend to share, as follows: (1) These sectors are the major contributors of waste load discharge in the district. (2) Moreover, they have the most pollutant discharge per unit of GDP, implying the lowest environmental and economic benefits in relation to water resource utilization, and (3) at the same time, they feature lower removal costs and operational challenges. Therefore, the pollution sources with the above-mentioned characteristics should be the top priorities in the reduction of surplus waste load discharge.

It is also noteworthy that most previous studies focused primarily on the WLA of removals among PS pollution. Conversely, our results highlight the industrial pollution source as the last option for reduction from an environmental-economic benefit perspective. Instead, the often overlooked types, such as agricultural NPS and domestic sources, deserve more attention, especially in extensive rural areas.

Jinping town was selected as a typical example to demonstrate the optimal allocation of removals among pollution sources. Further efforts should include comparative analyses of districts with various

landscape features to explore the optimization of WLA at the pollution source scale, under different regional characteristics. The multi-scale and multi-sector WLA method developed in this study can provide an important reference for similar research in other watersheds, especially for extensive developing areas that are subjected to multi-source pollution and serious surface water deterioration.

**Author Contributions:** Conceptualization, Q.L. and J.Q.; Data curation, Q.L.; Formal analysis, Q.L.; Funding acquisition, Z.L. and J.Q.; Investigation, Q.L. and J.J.; Methodology, Q.L. and J.J.; Project administration, Z.L. and J.Q.; Software, Z.L. and J.Q.; Supervision, Z.L.; Validation, Q.L. and J.J.; Visualization, Q.L.; Writing—original draft, Q.L.; Writing—review & editing, C.J. and J.Q.

**Funding:** This research was funded by Asian-Pacific Network for Global Change Research, grant number ARCP2015-06CMY-Wu, and USDA project through the AgBioResearch at Michigan State University, grant number MICL02264.

**Acknowledgments:** The authors would like to specially thank the Ningbo Scientific Research and Design Institute of Environment Protection for providing the valuable census data of pollution sources. The socio-economic statistical data were provided by the township governments of Ningbo city. We also gratefully acknowledge the administrative support by the Ningbo Environmental Protection Bureau.

**Conflicts of Interest:** The authors declare no conflict of interest. The sponsors had no role in the design, execution, interpretation, or writing of the study.

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
