# Peer review of "A New Water Environmental Load and Allocation Modeling Framework at the Medium–Large Basin Scale"

_water, doi:10.3390/w11112398_

Round 1

Reviewer 1 Report

In the manuscript "A new water environmental load and allocation modeling framework at mediu-latge basin scale" the authors describe a new method to allocate pollution loads.

I believe the manuscript is of interest but some issues may be adressed prior to publication:

the results and discussion is too long, and sometimes dificult to follow for the reader; an effort should be made to improve this section for clarity; the role of the experts in the process should be clarified; the abstract has a very large introduction (about the background) that can be simplified; some revision regarding the English language is needed to improve the clarity of the manuscript.

I am attaching the commented version of the manuscript.

Author Response

Dear reviewer,

   We have completed the revision and supplement responses to each of the comments and suggestions that you put forward and uploaded the final state of the revised manuscript and cover letter to the attachment.

    We would like to express our great appreciation for your valuable comments and suggestions on our manuscript!

      Best wishes!

Reviewer 2 Report

Reviewer is impressed by the amount of work, investigations and idea which have been done by the Authors. The idea seems to be novel with maybe a average impact in science but the great influence in practical water/wastewater managing. The paper shows clearly the idea and the course to create results with their discussion. Reviewer hasn't found any mistakes and incomprehensions. In reviewer's opinion the manuscript is suitable for publish in Water.

Author Response

Dear reviewer,

  I would like to express on behalf of my co-authors great appreciation for reviewing the article in your busy time and getting your recognition to our research results!

    Best wishes!

Round 2

Reviewer 1 Report

The authors took into consideration the comments a did a good work in the revised version of the manuscript. In my opinion the manuscript is can now be accepted for publication.